

# Combining multisensor images and social network data to assess the area flooded by a hurricane event

Rafael Hernández-Guzmán[1] and Arturo Ruiz-Luna[2]

[1] CONAHCYT - Instituto de Investigaciones sobre los Recursos Naturales, Universidad Michoacana de San Nicolás de Hidalgo, Morelia, Michoacán, Mexico
[2] Manejo Ambiental, Centro de Investigación en Alimentación y Desarrollo (CIAD–Mazatlán), Mazatlán, Sinaloa, Mexico

Corresponding author
Rafael Hernández-Guzmán,
rafael.hernandez@umich.mx

## ABSTRACT

In this study, multisensor remote sensing datasets were used to characterize the land use and land covers (LULC) flooded by Hurricane Willa which made landfall on October 24, 2018. The landscape characterization was done using an unsupervised K-means algorithm of a cloud-free Sentinel-2 MultiSpectral Instrument (MSI) image, acquired during the dry season before Hurricane Willa. A flood map was derived using the histogram thresholding technique over a Synthetic Aperture Radar (SAR) Sentinel-1 C-band and combined with a flood map derived from a Sentinel-2 MSI image. Both, the Sentinel-1 and Sentinel-2 images were obtained after Willa landfall. While the LULC map reached an accuracy of 92%, validated using data collected during field surveys, the flood map achieved 90% overall accuracy, validated using locations extracted from social network data, that were manually georeferenced. The agriculture class was the dominant land use (about 2,624 km$^2$), followed by deciduous forest (1,591 km$^2$) and sub-perennial forest (1,317 km$^2$). About 1,608 km$^2$ represents the permanent wetlands (mangrove, salt marsh, lagoon and estuaries, and littoral classes), but only 489 km$^2$ of this area belongs to aquatic surfaces (lagoons and estuaries). The flooded area was 1,225 km$^2$, with the agricultural class as the most impacted (735 km$^2$). Our analysis detected the saltmarsh class occupied 541 km$^2$ in the LULC map, and around 328 km$^2$ were flooded during Hurricane Willa. Since the water flow receded relatively quickly, obtaining representative imagery to assess the flood event was a challenge. Still, the high overall accuracies obtained in this study allow us to assume that the outputs are reliable and can be used in the implementation of effective strategies for the protection, restoration, and management of wetlands. In addition, they will improve the capacity of local governments and residents of Marismas Nacionales to make informed decisions for the protection of vulnerable areas to the different threats derived from climate change.

## INTRODUCTION

Although floods are natural phenomena that occasionally occur in all rivers and drainage systems, they are also a source of major disasters that affect many countries in the world (*Zhang et al., 2020*; *Adedeji et al., 2021*). Besides damaging lives, natural resources, and

the environment, reducing biodiversity and injuring wetland areas, also causes economic losses (*Zhang et al., 2015*; *Bai et al., 2021*). To reduce the negative effects, preventive disaster response management requires tools, such as potential flooding maps, to identify the possible extent of the event, and the expected damage to infrastructure and assets (*Caballero, Ruiz & Navarro, 2019*; *Tripathy & Malladi, 2022*).

The use of traditional methods such as ground surveys in mapping the extent of floods is time-consuming (*Brivio et al., 2002*; *Rahman & Thakur, 2018*). During flood events, field data are restricted to an inadequate number and distribution of point locations and transects that rarely coincide with the inundation. Its collection is expensive due to the logistical problems associated with access to the flooded area when the communication and transportation facilities are damaged or disrupted for a long time due to heavy rainfall. In this scenario, traditional methods account only for small scale results, and conducting a field survey when the phenomenon is widespread can be quite difficult. In addition, these field surveys require enormous resources such as skilled individuals, and financial and computational resources (*Singh & Kansal, 2022*).

Currently, the only way to monitor the catastrophic impacts of flooding across large areas and over time, is through satellite images because they offer frequent observations and historical archives (*Caballero, Ruiz & Navarro, 2019*; *Tulbure et al., 2022*). Optical satellite data such as the Moderate Resolution Imaging Spectroradiometer (MODIS) imagery (*Coltin et al., 2016*; *Fayne et al., 2017*) at high temporal resolution (daily) but a coarser resolution (250 m), Landsat (*Mehmood, Conway & Perera, 2021*; *Sivanpillai et al., 2021*) at a higher spatial resolution (30 m) but lower temporal resolution (16-day), and more recently, the European Space Agency (ESA) Copernicus Sentinel-2 (at 10 m spatial resolution and five days temporal resolution), have been mostly used for flood mapping at a catchment scale (*Caballero, Ruiz & Navarro, 2019*; *Singh & Kansal, 2022*).

Confidence in these remotely sensed data relies on the fact that they are systematically acquired, easily accessible, and available globally at little or no cost (*Tulbure et al., 2022*). However, optical satellite images have a technical limitation in delineating the flood extent when cloud cover is extensive (common during flooding time) restricting data collection (*Liang & Liu, 2020*; *Psomiadis, Diakakis & Soulis, 2020*; *McCormack, Campanyà & Naughton, 2022*). Therefore, Synthetic Aperture Radar (SAR) data are used to overcome this limitation.

While the use of SAR imagery for flood mapping is well established in the scientific literature (*Matgen et al., 2011*; *Munuwar, Hammad & Waller, 2022*; *Tarpanelli, Mondini & Camici, 2022*), it was not until 2014, with the launch of the ESA Copernicus Sentinel 1 satellite, with a C-band SAR, that its use became increasingly widespread (*Sharifi, 2020*; *Tulbure et al., 2022*). The Sentinel-1 satellites, and in general all SAR systems, are cloud penetrating and have a day-and-night capability, enabling them to acquire imagery regardless of the weather and facilitating the monitoring and mapping of floods, sea ice, oil spills, land-surface motion, and more (*ASF, 2023*).

Nowadays, SAR data are frequently used in combination with optical satellite data for mapping floods (*Azizian & Brocca, 2020*; *DeVries et al., 2020*; *Psomiadis, Diakakis & Soulis, 2020*). Also, to enhance flood predictions, digital elevation models (DEMs) or

digital terrain models (DTMs) are incorporated within hydraulic modeling to produce flood extent and depth maps (*Scotti, Giannini & Cioffi, 2020*; *Sharifi, 2020*; *Levin & Phinn, 2022*). However, these high-quality elevation datasets are unavailable in most parts of the world (*Azizian & Brocca, 2020*), restricting the use of these methodologies especially in data-limited regions. This is particularly true in Mexico where, despite the frequency of flood events on the western coast of Mexico during the rainy season (July–November), most of the aforementioned approaches have been of limited use, with few exemptions in Sinaloa (*Peinado-Guevara et al., 2022*) and Nayarit (*Hernández-Guzmán et al., 2016*), where the study area is located.

This study aims to document the different land use and land covers affected by the floods caused by Hurricane Willa on October 24, 2018. Although this hurricane made landfall as a Category 3 hurricane on the Saffir-Simpson scale, with winds of up to 195 kmph in the state of Sinaloa, Mexico, the most significant impacts were caused in the coastal zone of the state of Nayarit, particularly with the overflows of the Acaponeta and San Pedro rivers. It combines optical satellite image and SAR data analysis to map the flooding extent, and social network data to evaluate its accuracy. These data were valuable tools to assess the flooded area by Hurricane Willa.

## MATERIALS & METHODS

### Study area

The study area is in the Northwest of Mexico in the state of Nayarit between 21°35′ and 22°38′ North latitude and 105°00′ and 106°00′ West longitude (Fig. 1). It covers a total around 8,200 km². In this area is the Marismas Nacionales Biosphere Reserve (MNBR), a refuge and habitat area for both local and migratory bird species that houses the most extensive mangrove complex in the Mexican Pacific and integrates different types of coastal wetlands where the lagoons, saltmarshes, and mangroves stand out. The mangroves of the zone are considered the most developed mangroves in the eastern Pacific (*Muro-Torres et al., 2020*). Currently, MNBR is one of the most threatened and deteriorated wetlands zones and their degradation continues due to the cumulative effect of multiple threats that have impacted changes in hydrological patterns, sedimentation, and topography.

This region presents a summer rainfall pattern, with more than 80% of the annual precipitation falling from June to October (the rainy season). Even when there are six rivers in the study area, only three of them present permanent water discharge (Acaponeta, San Pedro, and Santiago rivers). The Acaponeta and San Pedro rivers are significant because they are the only watercourses in the region, discharging directly into the lagoon system and dams free.

Two data sets were used to map the area perturbed by the flooding produced by the Hurricane Willa effects; a dataset consisting of optical images to make a landscape characterization during May 2018, considered as the reference image, and a second dataset composed of multispectral and SAR images, recorded on dates after the flood event.

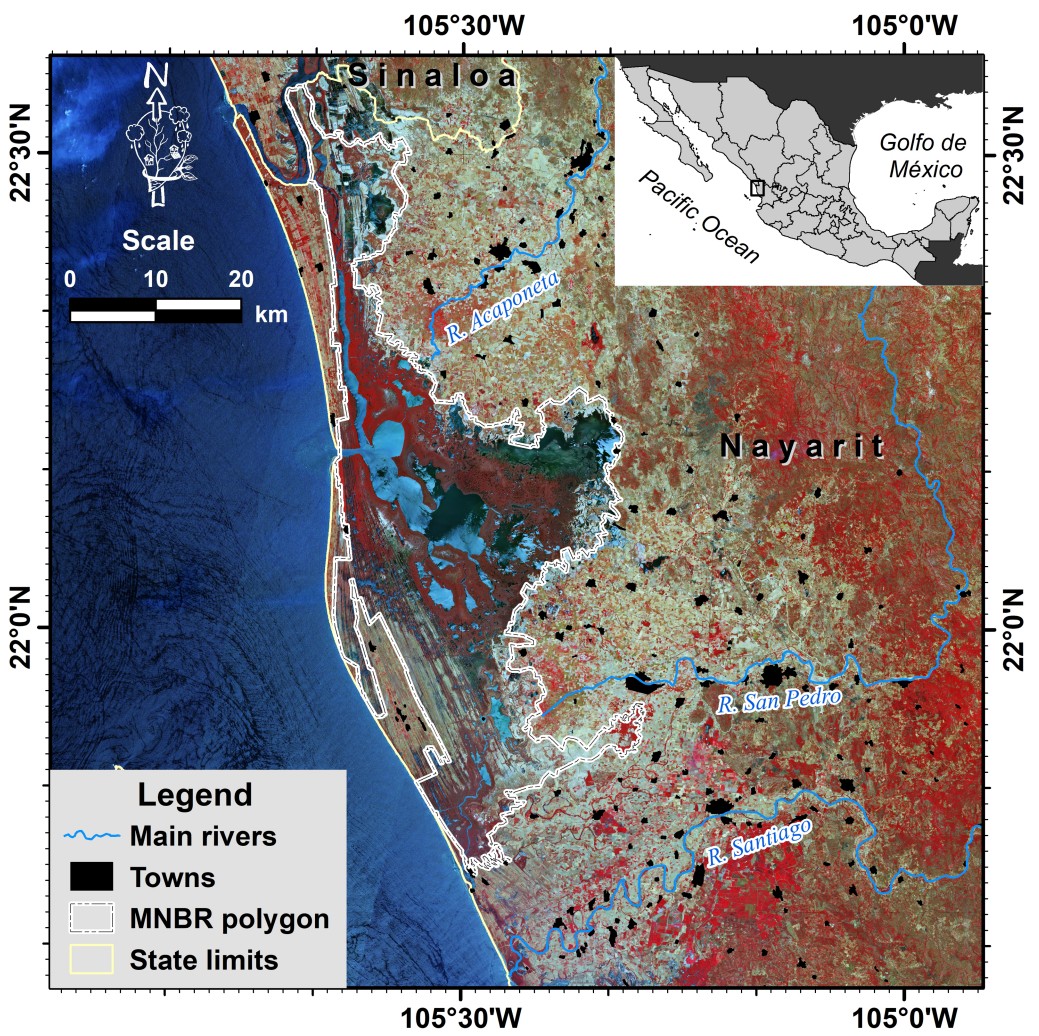

**Figure 1** **Geographic location of the study area in the state of Nayarit, Mexico.** Map source credit: Images were processed from Copernicus Sentinel-2 data (2018), ©European Space Agency-ESA.

## Landscape characterization

A cloud-free Sentinel-2 MultiSpectral Instrument (MSI) image (Tile T13QDE from May 21, 2018) acquired during the dry season previous to the Hurricane Willa landfall was used for the landscape characterization. This scene was independently classified into eight informational classes, using an unsupervised K-means algorithm. This algorithm uses a K-means clustering technique to divide an n-dimensional image into K unique clusters. In the first step, the algorithm locates the K-means (centroids) and then maps the pixels to the nearest mean using the Euclidean distance. In a second step, these means are updated, and the pixels are reassigned to a class. The process is repeated until the means are fixed (*Eastman, 2016*). In this analysis, all the pixels of the B2 (blue), B3 (green), B4 (red), and B8 (near-infrared) bands, with 10-m resolution, were formerly assigned to 50 spectral classes, which were later reclassified into the following informational classes:

aquatic surfaces (estuaries, lagoons, rivers), mangroves, saltmarshes, agricultural lands, sub-perennial forest, tropical deciduous forest, grassland and littoral. Urban areas or towns, and aquaculture classes were also included, but they were digitized on-screen over false-color compositions of Sentinel-2 images.

## Flood characterization

A Sentinel-1 C-band, recorded on October 26, two days after Hurricane Willa made landfall, corresponding to the Path-Frame 85-518, VV + VH Polarization, and a Sentinel-2 multispectral image (October 28, 2018), were analyzed to determine the extent of the flooding. The Sentinel-2 image was classified as described above, but only considering the flooded area.

Regarding the Sentinel-1 image, a radiometric calibration was performed by selecting the VV polarization and then the speckle noise (salt and pepper texture) was reduced by applying a spatial filtering type Lee with a window size of 3×3. The resulting image was co-registered with the Range-Doppler Terrain Correction algorithm using the DEMs from the Shuttle Radar Topography Mission (SRTM) and a bilinear interpolation as the resampling method. The binary map with flooded and non-flooded areas was generated by applying an estimated threshold value from the bimodal histogram of the image (*Bekele et al., 2022*; *Jesudasan, Subbarayan & Devanantham, 2022*). This method relies on the image having a bimodal histogram for proper separation, which means that the contrast in values between the flooded and non-flooded pixels should be as high as possible.

Finally, to detect the flooded surface, boolean images were created with the pixels detected for this informational class in both the Sentinel-1 SAR and sentinel-2 MSI post-event images. These images were added in a single cover class and later superimposed on the land use and land cover map to generate statistics for the land use and land covers affected.

All of the images were downloaded from the European Space Agency (ESA) official website Copernicus Open Access Hub (https://scihub.copernicus.eu/dhus/). The Sentinel-2 MSI images were processed using the TerrSet Software, while the Sentinel-1 SAR image was processed with the Sentinel Application Platform (SNAP) software from the ESA (http://step.esa.int/main/download/snap-download/).

## Accuracy assessment

The accuracy of the land use and land cover map was evaluated using an error matrix constructed with ground control points collected with a Garmin Montana 680 GPS unit during field surveys in 2019 combined with points extracted from Google Earth by an independent analyst. From this error matrix, indicators of classification accuracy such as overall accuracy, user's accuracy (UA), the producer's accuracy (PA), and the Kappa coefficient of agreement ([K]) were obtained (*Congalton & Green, 2019*).

To assess the overall accuracy of the flood map, photos from the social network Twitter (now called X) posted under several hashtags (*i.e.,* #HuracánWilla, #FuerzaNayarit, #MexicoTeNecesitamos, #Acaponeta, #Tecuala, #Tuxpan, #Ruiz, #Nayarit, #Willa) after the hurricane made landfall were downloaded. Our efforts were mainly focused on searching

photos posted by national and regional agencies and then those photos shared by Twitter users. Taking into account that the photos downloaded had no information associated in the metadata regarding their location (*i.e.,* coordinates), they were sorted according to the locality label (*e.g.*, #Tuxpan, #Acaponeta, *etc.*) considering the text associated with the photo. Each photo was imported to and manually georeferenced in Google Earth using the Google Street View (GSV) application through Google Maps as ancillary information. The x- and y- coordinates of the nearest geo-location were extracted from these photos. GSV, an online virtual imaging application, is a feature of Google Maps/Google Earth that enables users to view and navigate through 360- degree street-level images. In addition, our flood map was compared with those produced by the Copernicus Emergency Management Service (CEMS) Rapid Mapping. The CEMS uses a similar approach as in this study and provides mapping services whose products can be used to monitor and forecast natural disasters like floods, forest fires, or droughts (*Scalia, Francalanci & Pernici, 2022*).

## RESULTS

The land use and land cover map obtained by classifying a Sentinel-2 image produced an overall accuracy of 92% and a Kappa coefficient of 0.91. The user's accuracies ranged from 79% (agriculture class) to 100% (littoral and mangrove classes), while the producer's accuracies ranged from 75% (grassland class) to 98% (agriculture and aquatic surface classes). Additionally, based on 60 georeferenced points extracted from the social network Twitter photos, the overall accuracies were 23%, 88%, and 90% for the flooded area map produced by Sentinel-1, the Sentinel-2 map, and the map generated by the consensus between both sensors, respectively.

The landscape characterization, previous to the flood, indicates that the study area is mainly composed of upland covers (grasslands, deciduous and sub-perennial forests), which together occupy around 45.6% (3,740 km$^2$), mostly distributed to the west of the study area. The second largest class was agriculture (2,624 km$^2$), consisting of exposed soils (arable land) and standing crops, representing approximately 32% of the total area, distributed from north to south in areas with gentle slopes. The aquatic surfaces (estuaries, lagoons, and rivers) constituted the permanently inundated area which, together with the saltmarsh class, occupied ∼13% (1,030 km$^2$). Finally, the mangrove class occupies an extension of about 570 km$^2$ (∼7%). Fig. 2 shows the total estimated area (km$^2$) by land use and land cover, while Fig. 3 shows their spatial distribution in the study area.

Regarding the Hurricane Willa effects, excluding the aquatic surfaces (489.3 km$^2$), the Sentinel-1 sensor detected an inundated area of 603 km$^2$ (Fig. S1A), while the Sentinel-2 sensor produced almost twice this area with 1,165 km$^2$ (Fig. S1B). After adding both images into one, the consensus map outputs a flooded area of around 1,225 km$^2$, with only 544 km$^2$ detected coincidentally by both sensors.

The flooded areas were mainly distributed in five large zones. To the north, and associated with the Cañas and Acaponeta rivers, there were two main areas: (1) Close to the limit between Sinaloa and Nayarit states, characterized by saltmarshes and aquaculture ponds; (2) the Acaponeta river mouth, mainly occupied by agricultural land. Additionally, to the

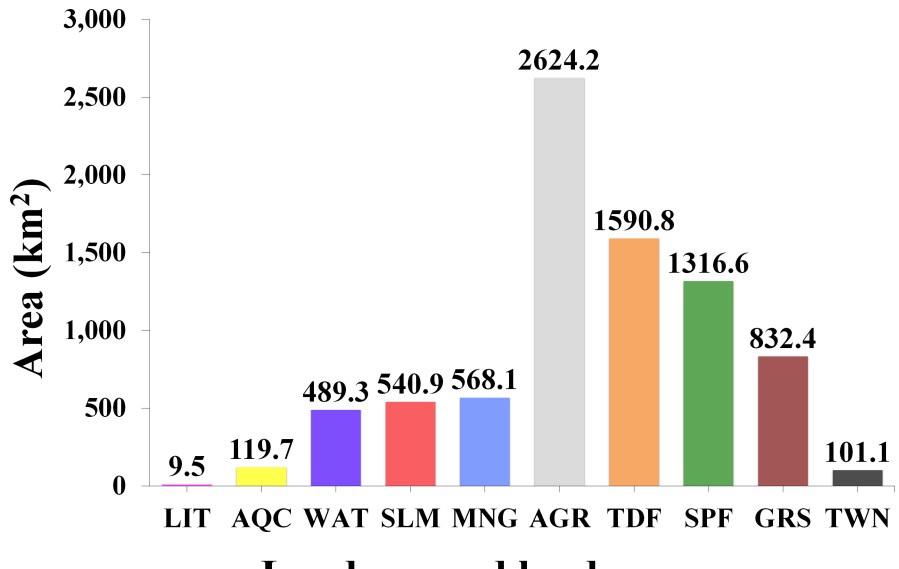

**Figure 2** **Total estimated area (km$^2$) by land use and land cover class for the year 2018 obtained through the unsupervised classification of a Sentinel-2 image at 10 m of spatial resolution.** Littoral (LIT); Aquaculture (AQC); Aquatic surfaces (WAT); Saltmarsh (SLM); Mangrove (MNG); Agriculture (AGR); Tropical deciduous forest (TDF); Sub-perennial forest (SPF); Grassland (GRS); Towns (TWN).

middle and south of the study area, three other flood zones were noticeable; (3) the central region of the study area, occupied by saltmarshes and dead and recovery mangroves; (4) the western fringe of the coastal zone of the study area, occupied by the mangrove and saltmarsh classes; and (5) the San Pedro river mouth, where agricultural lands are the main flooded areas (Fig. 4).

Considering the previous characterization, about 60% of the flooded area occurred on agricultural land (735 km$^2$), followed by the saltmarsh and aquaculture classes representing 27% (328 km$^2$) and 8% (99 km$^2$) of the surface detected as flooded, respectively (Fig. 5).

Finally, although towns represent less than 1% of the flooded area (11 km$^2$), they were also highly affected. Particularly Pajaritos, a small town located to the north of the study area, was devastated, while Tuxpan and El Mezcal, were inundated by the overflow of the San Pedro river, leaving numerous families homeless, limiting their livelihoods and damaging infrastructure (Fig. 6).

## DISCUSSION

A multi-sensor remote sensing approach was used to characterize the land use and land covers (LULC) flooded by Hurricane Willa in October 2018 that impacted the coastal zone of northwest Nayarit, Mexico where the Marismas Nacionales Biosphere Reserve (MNBR) is located. While the LULC map reached an accuracy of 92%, validated using data collected during field surveys, the flood map achieved 90% in the overall accuracy validated using locations extracted from photos shared on the social network Twitter. These overall accuracies allow the presumption that our results are reliable and in line

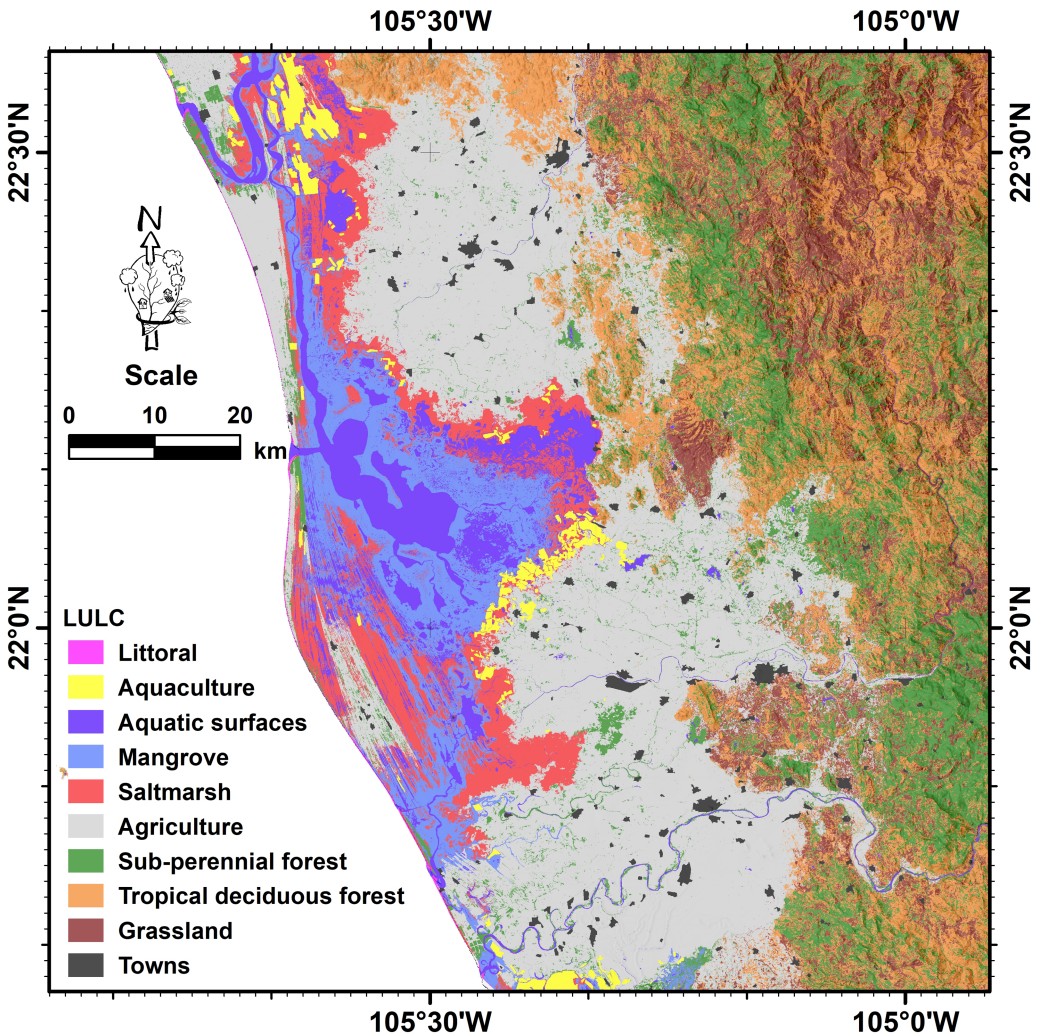

**Figure 3** **Spatial distribution of the land use and land cover classes for the year 2018 obtained through the unsupervised classification of a Sentinel-2 image at 10 m of spatial resolution.** Map credit: Images were processed from Copernicus Sentinel-2 data (2018). ©European Space Agency-ESA.

with what is expected of remote sensing accuracy assessments. The agriculture class was the dominant land use (about 2,624 km$^2$) but the most impacted by flooding (735 km$^2$). The approach used in this study detected which of the permanent wetlands (mangrove, salt marsh, and littoral classes) within the MNBR were the most impacted. Regarding this, from the 541 km$^2$ of the saltmarsh class, around 328 km$^2$ were under the water during Hurricane Willa.

The MNBR integrates a heterogeneous complex of wetlands characterized by a series of beach cords, parallel to the coastline, originating from coastal accretion processes (*Curray, Emmel & Crampton, 1969*). In addition to the beach ridges, which constitute the dunes, this complex includes lagoons, mangroves, and swamps, among other coastal wetlands, highly

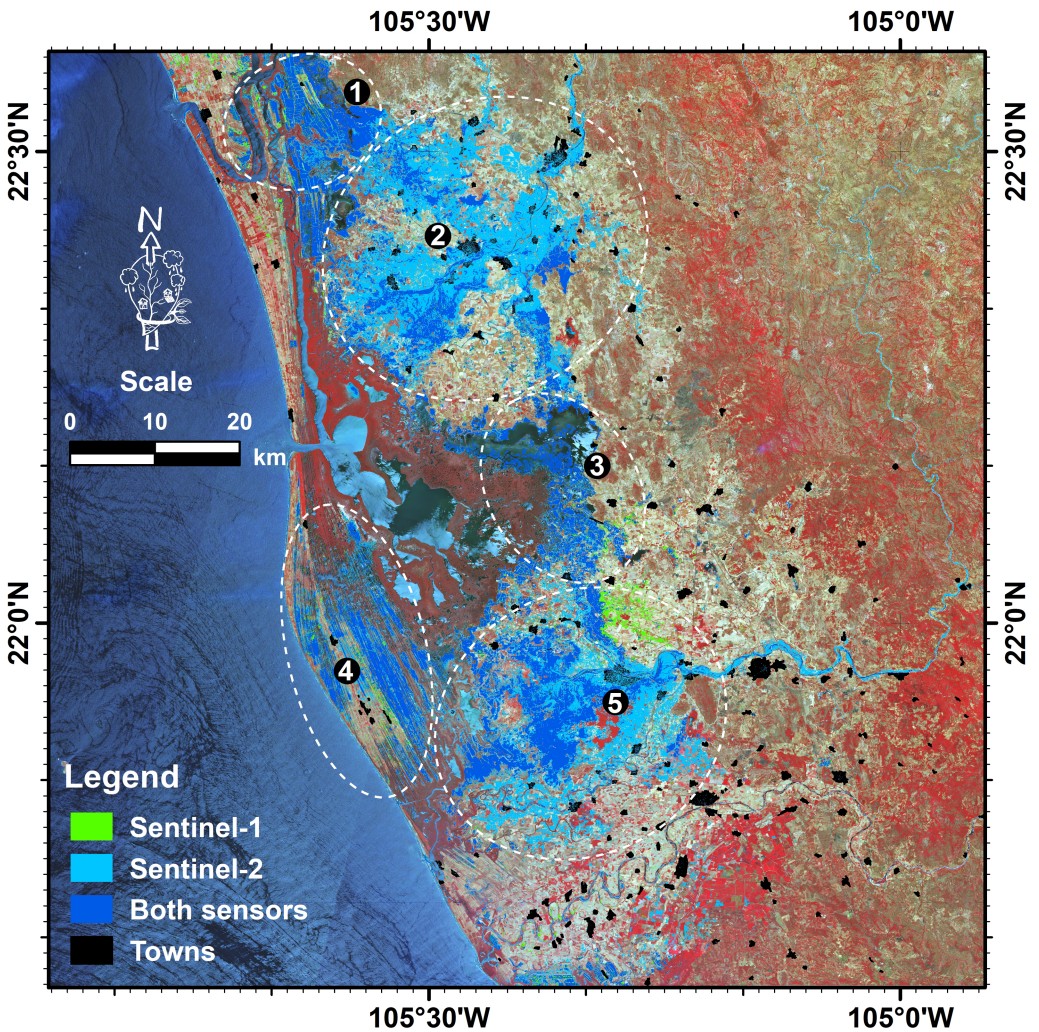

**Figure 4** **Spatial distribution of the flooded area by the time of satellite overpass.** Five large zones are identified: (1) The northern part of the Nayarit state; (2) the Acaponeta river mouth; (3) the central part of the study area; (4) the coastal zone at the central part of the study area; and (5) the San Pedro river mouth. Map credit: Images were processed from Copernicus Sentinel-1 and Sentinel-2 data (2018). ©European Space Agency-ESA.

influenced by a semi-diurnal tidal regime. This particular geomorphology contributes to periodical floods along the system.

Despite the broad spectrum of studies on the MNBR, the extent and condition of some ecosystems have been evaluated through different methodological schemes (*Lithgow, Lanza & Silva, 2019*; *Vizcaya-Martínez et al., 2022*), while few studies have included hydrological approaches (*Hernández-Guzmán et al., 2016*; *Salinas-Rodríguez et al., 2018*). In particular, *Hernández-Guzmán et al. (2016)* combined hydrological models and remote sensing data to evaluate the spatial dynamics of flood pulses in the area, recognizing that floods occur recurrently, associated with seasonal precipitation and runoff patterns upland, but are also promoted by tropical storms and hurricanes, like Hurricane Willa.

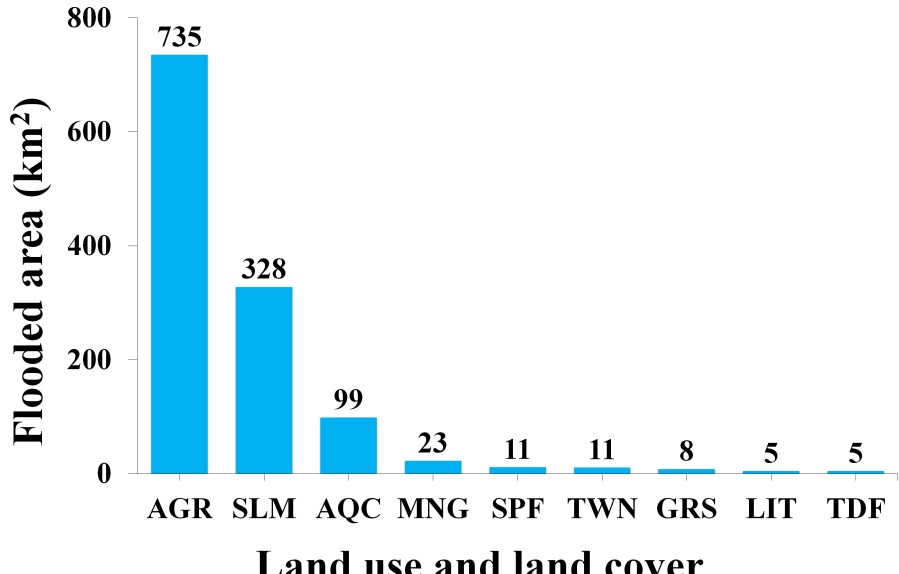

**Figure 5** **Total estimated area (km²) by land use and land cover class flooded during Hurricane Willa.**
Agriculture (AGR); Saltmarsh (SLM); Aquaculture (AQC); Mangrove (MNG); Sub-perennial forest (SPF);
Towns (TWN); Grassland (GRS); Littoral (LIT); Tropical deciduous forest (TDF).

Notwithstanding the flooded areas are now mapped using sophisticated deep learning (*Bai et al., 2021*; *Katiyar, Tamkuan & Nagai, 2021*) and machine learning algorithms (*Uddin, Matin & Meyer, 2019*; *Soria-Ruiz et al., 2022*), some researchers acknowledge that the histogram thresholding technique employed in this study is the simplest and most common procedure for floods mapping from SAR data. This technique is a fast, reliable, and computationally less time-consuming method (*Liang & Liu, 2020*; *Levin & Phinn, 2022*; *Rossi et al., 2023*). However, our SAR-based flood map generated using this technique, achieved an overall accuracy of 23%, as validated through social network downloaded photos. The flood events occurred at a regional level, mainly affecting a large spatial area along the lower sections of the Acaponeta and San Pedro rivers. The observed discrepancy (low accuracy) between flood maps derived from SAR images and the actual flooded lands can be attributed to the time lag between the flood peak and the satellite overpass (*Brivio et al., 2002*). To address this limitation, a proposed approach involves estimating the flooded area during the peak time by integrating SAR imagery with Digital Elevation Models (DEM) (*Brivio et al., 2002*), as the depth of the water in flooded areas can be estimated and correct some misclassified pixels (*Levin & Phinn, 2022*; *Nhangumbe, Nascetti & Ban, 2023*). However, as discussed by *Muhadi et al. (2020)*, although aerial photographs or globally available DEMs such as Advanced Spaceborne Thermal Emission and Reflection Radiometer (ASTER) and the Shuttle Radar Topography Mission (SRTM) are commonly used, these models lead to low accuracy of flood prediction due to the significant effect of low-accuracy DEMs. Besides, until now high-quality elevation datasets are unavailable in most parts of the world (*Azizian & Brocca, 2020*).

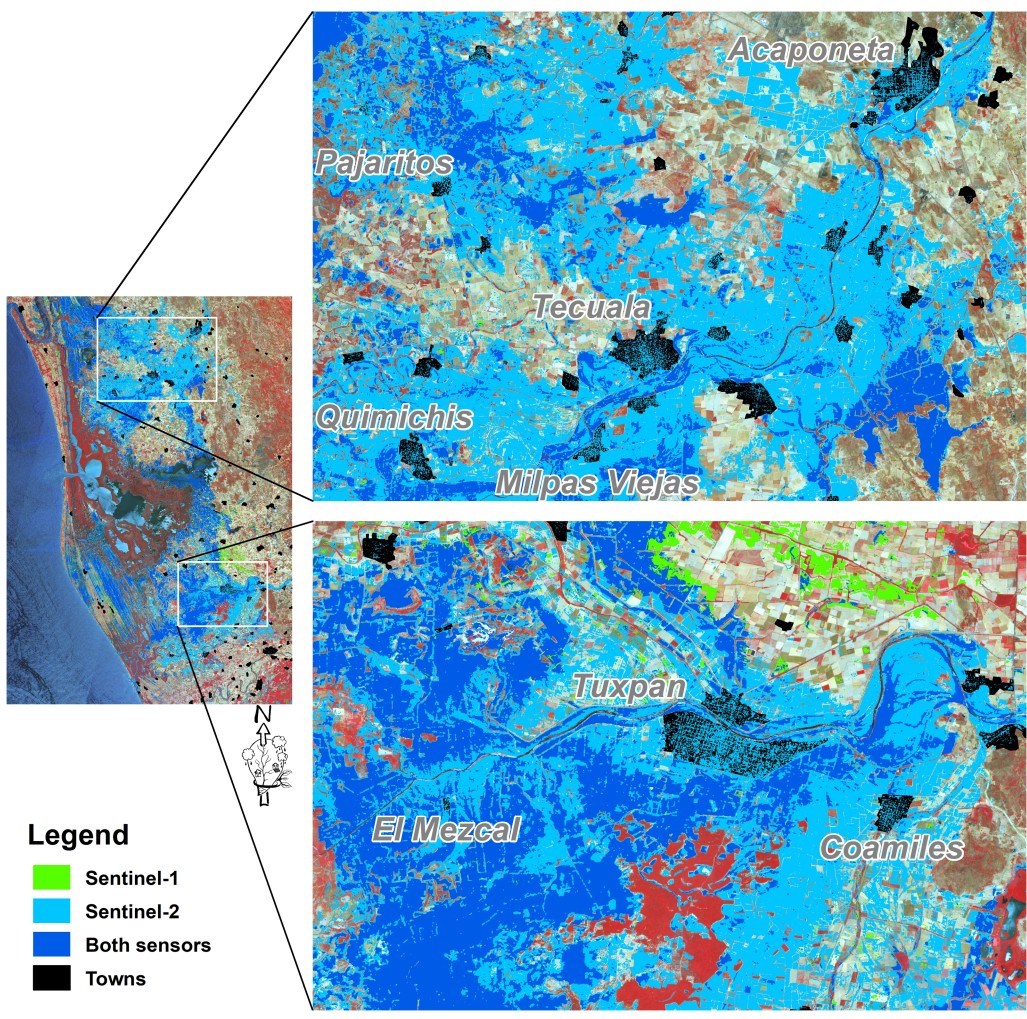

**Figure 6 Spatial distribution of the flooded area with zoomed insets for selected areas.** The upper and lower insets show the Acaponeta and San Pedro river sections, respectively. Map credit: Images were processed from Copernicus Sentinel-1 and Sentinel-2 data (2018). ©European Space Agency-ESA.

As stated before, although the Sentinel-1 and Sentinel-2 data are freely available through the Internet, a shortcoming will always be the time delay in their availability after a disaster event occurs (*Nhangumbe, Nascetti & Ban, 2023*). In this study, a Sentinel-1 image was acquired when Hurricane Willa made landfall (Fig. S2A), but the flood was visible just two days later (Fig. S2B). There is a difference of four days between the hurricane landfall and the Sentinel-2 image acquisition, considered here as the time of maximum flood (Fig. S3B). This later map was produced with 88% overall accuracy, and promptly, some badly affected towns were identified in the study area (*i.e.*, Pajaritos, Tuxpan, El Mezcal). The increase in accuracy was possible because water-saturated soils were identified in the multispectral image and considered flooded areas. After all, the limit between these two classes was not always evident in some parts of the study area.

Another aspect to consider is that almost all the flood water was completely drained by the time of the following Sentinel-1 and Sentinel-2 overpasses (Figs. S2C and S3C). As discussed by *Tarpanelli, Mondini & Camici (2022)* and *Zhang et al. (2015)*, flash floods are difficult to characterize due to their high intensity and sudden onset. Most of the satellite fails when flood inundations have a duration shorter than one week. This situation can be explained by the soil types in the lower sections of the Acaponeta and San Pedro rivers (Cambisol and Fluvisol, respectively). These soils have moderately fine to fine texture with a layer that impedes the downward movement of water that favored flooding after the ground was saturated but promptly drained and only detected as water-saturated soils during the November 2nd satellite overpass (Fig. S3C).

Some researchers recognize technical inconveniences of the optical sensors as the revisit time and their limitations due to cloudiness, a typical condition during the rainy season, when there is a greater probability of flooding (*Hernández-Guzmán et al., 2016*; *Adedeji et al., 2021*; *Tarpanelli, Mondini & Camici, 2022*), suggesting the use of SAR data to elude or reduce them. However, our findings highlight the importance of using Sentinel-2 data for flood mapping compared with the Sentinel-1 data. Therefore, a common complementary approach is to use different SAR and optical datasets together because their combined use provides better results than using individual sensors (*Nhangumbe, Nascetti & Ban, 2023*). In this regard, the thresholding method to the SAR image combined with the analysis of a multispectral image and the comparison with pre-event imagery (containing information on the permanent water bodies) produced a flood map with a relatively high accuracy (90%) which is comparable to those obtained with complex methods (*Pedzisai et al., 2023*; *Tupas et al., 2023*).

After a visual inspection, most omission errors were detected in the adjacency or peripheral of the town class since these validation points collected from Twitter photos were classified as non-flooded. Notwithstanding this, our results suggest an accurate flood map for the urban areas. *Scotti, Giannini & Cioffi (2020)* reported this situation by reconstructing the flooded urban areas associated with Hurricane Harvey in Houston, TX. These researchers used an integrated approach that combines satellite images of flooded areas, hydraulic models, and markers from social media being able to obtain an accurate flooding map of the event, even in urban areas where satellite data fail to provide information.

The Government of Mexico stated that Hurricane Willa produced heavy rainfall with two main consequences in the days after its landfall: The first was the flooding, derived from the torrential rains that hit the state of Nayarit. The second one, and more serious, was the river flood resulting from the overflowing of the Acaponeta and San Pedro rivers (*Navarro-Quintero, 2018*). Our findings support this statement as illustrated in Fig. 4, where flooding areas were not limited to river-fed flooding only (areas 2 and 5) but also to rain-fed flooding (areas 1, 3, and 4).

Temporary flooding due to rainfall may occur locally in the agricultural areas adjacent to the wetlands of the San Pedro River (*Hernández-Guzmán et al., 2016*). The overbank river flow occurs because streams both collect water from runoff generated upland and from rainfall falling directly on them. Thus, the areas adjacent to rivers and their floodplains have
the biggest flood susceptibility. However, significant damage to homes and infrastructure occurred during Hurricane Willa, with more than 100,000 people having been displaced from their homes due to floodwaters (*Brennan, 2019*). The Government of Mexico reported that about 600 km$^2$ of the agricultural surface were flooded, with rice and corn fields as the most affected crops. Our results are 135 km$^2$ higher than the official figures. This is probably because the official figures correspond to a preliminary estimate, assuming our results are more accurate because the estimates were obtained through the analysis of satellite imagery.

It is important to mention that some limitations of our flood model validation approach were detected. On the one hand, during Hurricane Willa, the Copernicus Emergency Management Service (CEMS) Rapid Mapping was activated (activation EMSR328 Tropical Cyclone Willa in Nayarit and Sinaloa, Mexico), producing five rapid maps for the event. Despite the disclaimer "the thematic accuracy might be lower in urban and forested areas due to inherent limitations of the SAR analysis technique" stated in the map description, the provided map products are considered accurate in some regions (*Scalia, Francalanci & Pernici, 2022*). For example, *Nhangumbe, Nascetti & Ban (2023)* compared automatically mapped flooded areas with CEMS maps obtaining an overall accuracy of about 88%, showing a substantial agreement with the reference data. Regarding our study area, products of the EMSR328 show a lot of information gaps. The CEMS maps reported about 202 km$^2$ as the total flooded area, from which 177 km$^2$ (88%) agree with our estimated area using our approach. One point to remark is that CEMS Rapid Mapping can only be triggered by or through an Authorized User, which includes National Focal Points in the EU Member States and countries participating in the Copernicus program. This confers a weakness to the CEMS products, as the activation was only for the Acaponeta river and southern Sinaloa. Thus, multiple flooded areas were missed in the maps generated by the CEMS, as the San Pedro River was not on the Activation list.

On the other hand, the validation of the flooding map using Twitter photos was a challenge due to the lack of geolocated data. This procedure was conducted by comparing the computed data with validation points extracted from a limited number of Twitter photos published by users as well as national and regional agencies that were manually georeferenced. As discussed by *Ning et al. (2020)*, manually dealing with massive social media posts is inefficient. Besides, all posts containing flood photos might be omitted if they are not tagged with a metadata tag related to the disaster event in the text. Therefore, an insufficient number of tweets collected could become a limitation for an accurate flood delineation. This situation has already been reported by *Sadiq et al. (2022)*, who suggest that to remedy the data scarcity issue, there is a direct need for a geolocation inference algorithm that can geotag tweets using various metadata fields. Although a lot of work has been done in improving the tweet collection rate to obtain a much larger sample (*Gentry, 2022*) as well as the extraction of the post geolocation from Twitter metadata (*Scalia, Francalanci & Pernici, 2022*), this is usually available only in a small percentage of the posts. This situation can be exacerbated during extreme weather events, such as Hurricane Willa, which caused power failure and communication disruption.

## CONCLUSIONS

The thresholding method for SAR data performed very well in creating a binary classification of water and non-water regions. However, it was not suitable for mapping the flooded but drained area, as the derived SAR-based flood map shows only the flooded but not the water-saturated area. When water-saturated soils, detected in the multispectral image, were included, the flood map overall accuracy was improved. Since the time delay between the flood peak and the satellite overpass was identified as a limitation, the combination of satellite images acquired by passive and active sensors, is recommended for mapping flooded area, particularly for well-drained soil types, present in the study area. Therefore, similar results to those obtained here can be useful for validating hydrodynamic flood models and help disaster management agencies and other stakeholders coordinate prevention measures and recovery activities for future flood events.

Social network data played a crucial role in the validation process, but it is necessary to promote campaigns for community engagement in collecting ground control points for detailed flood characteristics (depth, location) as data gathered using electronic technologies (mobile devices) provide vital information for flood hazard monitoring. Social network users are encouraged to share their photos with a geotag to avoid an incorrect photo interpretation regarding the place where the photograph was taken, and the location of the tweet posted online.

Present results highlight the need for good flood products in Mexico that can be used in large spatial extent research as well as for validation. Since the Copernicus EMS Rapid Mapping service can only be triggered by or through an Authorized User, governmental agencies from Mexico are encouraged to be authorized users of the Copernicus EMS mapping to trigger the service in case of future disaster events. Finally, to avoid dependence on a third-party resource to validate flood maps and reduce errors, it is suggested to have at least one restricted area as a calibration area (*e.g.*, 100×100 m) that has no flood records during the experiment period.

## ACKNOWLEDGEMENTS

This study is part of the Cátedras CONAHCYT project (Project No. 148). The Sentinel-1 and Sentinel-2 images were obtained from the Copernicus Open Access Hub run by the European Space Agency.

### Funding

The authors received no funding for this work. The Coordination of Scientific Research (Universidad Michoacana de San Nicolás de Hidalgo) supported the publication fee. The funders had no role in study design, data collection and analysis, decision to publish, or preparation of the manuscript.

## Grant Disclosures

The following grant information was disclosed by the authors:
The Coordination of Scientific Research (Universidad Michoacana de San Nicolás de Hidalgo).

## Competing Interests

The authors declare there are no competing interests.

## Author Contributions

- Rafael Hernández-Guzmán conceived and designed the experiments, performed the experiments, analyzed the data, prepared figures and/or tables, authored or reviewed drafts of the article, and approved the final draft.
- Arturo Ruiz-Luna conceived and designed the experiments, analyzed the data, authored or reviewed drafts of the article, and approved the final draft.

## Data Availability

   The raw data is available in the Supplemental File.

## Supplemental Information

Supplemental information for this article can be found online at http://dx.doi.org/10.7717/peerj.17319#supplemental-information.

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
