# Peer review of "Combining multisensor images and social network data to assess the area flooded by a hurricane event"

_PeerJ, doi:10.7717/peerj.17319_

## Round 0.1 · original submission · Major Revisions

Dear Dr. Hernández-Guzmán,

In addition to the reviewers comments I would add the following:

Overall, the Discussion needs some improvements. The Discussion section should aim to summarize the main findings of the manuscript within the context of the broader scientific literature, and it should address any study limitations or results that may be in contrast with previously published research. The manuscript should provide practical and feasible policy recommendations based on the research results. It is suggested that the author emphasize this study's practical significance.

Kind regards,
Alban

·

Basic reporting

1. Basic reporting
a. There is not a clear linguistic flow of the manuscript. There must be a logical flow of the text as well as correct English spelling in some cases.
b. The manuscript has unnecessary to many references. They must be grouped and be at most 45 – 50 citations.
c. The title of the manuscript should be changed as there in no any association of social network in this work except of some photos which are not clear how are the associated with the research.

Experimental design

2. Experimental design
a. Not clear at all
b. Nothing is written about restriction of this method as it is done in the introduction part with the restrictions of other ones.
c. There should be at least one restricted area (very small one) like 100x100m as the calibration area/part of their experiment in order to check the accuracy or to do necessary corrections of their findings

Validity of the findings

3. Validity of the findings
a. The maps and table in the manuscript are well prepared
b. It is not possible to discuss the validity of the findings as authors by them self has not done it in their work.
c. It is not clear to be what is this manuscript real original contribution.
d. If they still will incorporate the social networks in their work, it must be clearly stated and combined with any/multi sensor images.
e. The abstract and conclusion part must be written based only in authors findings.

Additional comments

The flooding is major concern for all engineers and beyond. In this perspective every contribution is welcome in the academical field. However, every work must be clear and original in order to be published. I strongly believe authors will reorganize their manuscript in an acceptable form for publication

Reviewer 2 ·

Basic reporting

The authors use clear and professional English as used in scientific articles. Literature references are sufficient.

Experimental design

The study uses a clear study design combining remote sensing and mentions of the flood event in social media.

Validity of the findings

The study is an inventive combination of remote sensing and social media to produce more accurate regional findings. This work is useful and fits the scope of this journal. Conclusions are clear.

Reviewer 3 ·

Basic reporting

The manuscript by Hernández-Guzmán and Ruiz Luna proposes and describes a criterion to reconstruct post-event flood maps using multisensor remote sensing datasets and social network data. The latter were used to evaluate the accuracy of flood maps. The paper is well written and structured, the quality of figures is good. Objectives and methodology are clearly described, and results seem to support the proposed methodological approach

Experimental design

I found the approach proposed not particularly original or novel. The combined use of SAR and optical satellite imagines is rather common. Most emergency services such as for instance Copernicus Emergency Mapping Service (CEMS) routinary use a similar approach. Also, the validation of the methodology by social network data is not novel, being recently proposed and used by a number of authors.

Validity of the findings

Concerning the particularly good accuracy obtained (about 90%) by the authors I think that it depends on the specific characteristics of land cover of the flooding region and on the long-lasting flooding produced by the extreme events considered. I believe that both have reduced in the case examined the influence of the limitation of SAR and Optical satellite sensors. So, I think that in different conditions such limitations could more seriously affect the accuracy of flood maps. Thus, I don’t believe that the proposed approach could be considered of general validity.

Additional comments

In the paper the validation of the method by social network data is presented in a rather rough way and should be developed in more detail before Acceptance. However I think that, given the overall good quality of the manuscript and the interesting study case proposed, the manuscript could be considered for publication since in each case it presents an interesting applicative study case.

---

## Round 0.2 · Minor Revisions

The authors have addressed all of the reviewers' comments. Reviewer 2 did not reply to the invitation to review the revised manuscript, so I reviewed it myself. I see that the authors present the results only through some maps, but I did not see any quantitative assessment. So, I would like to see your findings presented quantitatively and measurably through graphs. Also, please check the legend of the maps if something is missing.

·

Basic reporting

The authors have done a satisfactory job in improving the previous manuscript. In general, the comments have been correctly addressed and solved and placed in the rewritten manuscript. The language used in writing the text and the logical flow of the treatment of the experiment is brought to a satisfactory state

Experimental design

The experiment itself is quite interesting and creates a new way in the literature for the treatment of floods and other events that have to do with flooding and its impact. The only concern, which has been asked to the authors in the previous comments, is the creation of a small surface in order to do the calibration. The authors have no opportunity since the event belongs to a past time. However, in the final manuscript, the calibration part of the experiment can be left as a suggestion

Validity of the findings

No comment

---

## Round 0.3 · accepted · Accept

Congratulations on your good work! I'm pleased to see that all my suggestions have been thoroughly considered and successfully implemented in the revised manuscript. The manuscript is ready for publication. Thank you for your valuable input.